
# Quantitative assessment of rainfall-induced landslide susceptibility in new urban area of Fengjie County, Three Gorges area, China

Haijia Wen [1,2], Yanyan Zhang [1,2], Guofan Duan [3], Hongmei Fu [1,2], Peng Xie [1,2] , Peng Zhou [4], Yong Yang [4]

[1] Key Laboratory of New Technology for Construction of Cities in Mountain Area(Chongqing University), Ministry of Education, Chongqing 400045, China

[2] School of Civil Engineering, Chongqing University, Chongqing 400045, China

[3] Chongqing Yuandao Construction Planning and Design Co., LTD, Chongqing 400041, China

[4] Chongqing Institute of Geology and Mineral Resources, Chongqing, 400042, China

*Correspondence to*: Yanyan Zhang (yyz@cqu.edu.cn)

**Abstract.** The objective of this study is to develop a methodology for quantifying rainfall-induced landslide susceptibility in a regional scale. Based on the combination of mechanical stability analysis and artificial neural network (ANN) and of Geographic Information Systems (GIS ) and detailed field investigation, the methodology was applied to the new urban area of Fengjie County in Northeastern Chongqing, China. According to the field investigation, an analysis sample database (ASD) pertaining to 6 slope stability influencing factors was built by means of uniform design method, and 30 samples for slope stability analysis were grouped. Then, safety factors of the sample groups were calculated by means of Geo-studio software concerning rainfall infiltration into slopes. To obtain overall slope stability analyses in the study area, the ANN was employed and the safety factors of the samples were utilized as training samples by ANN. Combining the trained ANN and survey data of the study area, the computation of safety factors under different rainfall were integrated and mapped within the GIS. The landslide susceptibility assessment indicates that slopes in more than a quarter of the study area are prone to landslides under rainstorm and severe rainstorm, however, slopes in the whole area under light rainfall, moderate rainfall and even heavy rainfall are relatively safer. Further, the results highlight the geological settings effect on landslide susceptibility as the high susceptibility zones are mainly distributed along the Yangtze River and its three branches, where the bank slopes are composed of fractured stratum, weak rocks and deposits. In good accordance with the rainfall-induced landslide events occurred in recent years and some findings in other literature about the study area, it is proved that the methodology presented in this paper could reasonably delineate landslide susceptibility under rainfall.

**Keywords:** Rainfall-induced landslide susceptibility; Three Gorges; Quantitative analysis; Geo-studio; Artificial neural network (ANN)

## 1 Introduction

Fengjie County is located in the Three Gorges region, known as an area of frequent landslides. Landslide hazards are increased in the Three Gorges area due to the construction of Three Gorges dam



(Bai et al., 2010). Attention has been attracted not only for landslide hazard assessment (e.g. Deng et al.,
2000; Wu et al., 2001; Fourniadis et al., 2007a; Wang and Li, 2012; Liu et al., 2013; Zhu et al., 2014)
but also for the impact of slope instability on ecosystems and socio-economic stability (Fourniadis et
al., 2007b).

5       In mountainous terrain, landslides are often triggered by rainfall (Dai and Lee, 2002a), which could

result in enormous property damage and loss of human life. In the case of landslide events, landslide
susceptibility assessment is the presentation of spatial distribution of existing and potential landslides
in an area (Guzzetti et al., 1999; Fell et al., 2008; Van Den Eeckhaut and Hervás, 2012) and it could
provide valuable assistance for hazards mitigation (Fall et al., 2006; Nefeslioglu et al., 2008). Studies
on rainfall-induced landslide have been conducted by various researchers around the world (e.g. Fourie,
1996; Crosta, 1998; Iverson, 2000; Dai et al., 2003; Rahardjo et al., 2005; Zhang et al., 2005; van
Wetsten et al., 2006; Crosta and Frattini, 2008; Castellanos Abella and van Westen, 2008; Wu and Chen,
2009; von Ruette et al., 2011; Giannecchini et al., 2012; Springman et al., 2013; Alvioli et al., 2014),
rainfall thresholds identification, rainfall infiltration analysis, stability analysis and landslide risk
assessment were implemented. Heuristic methods, statistical approaches, probabilistic and
deterministic models were employed concerning spatial and temporal characteristics as well as
site-specific slopes, regional scales and national scales were involved.
Based on the methods utilized to perform the landslide susceptibility, the most common
classification is to divide those methods into two types: qualitative methods and quantitative methods
(Aleotti and Chowdhury, 1999). The qualitative risk assessment based on heuristic approaches are
conducted in many countries (van Westen et al., 2006), for a regional analysis it is often useful (Ayalew
and Yamagishi, 2005), however, the qualitative methods are relied on the experience of experts and
hence partly subjectivity is accompanied. The quantitative methods, namely statistical analysis,
deterministic analysis and probabilistic approaches (Aleotti and Chowdhury, 1999), are based on
numerical calculations to figure out the relationship between influencing factors and landslides, thus
during the process of weight assignment subjectivity and bias could be minimized (Kanungo et al.,
2009). The appropriate choice of what type the methods are implemented depends on the type of
project, the availability of data, the criteria used to judge the degree of acceptable risk, etc. (Whitman,
2000). In addition, a detailed knowledge and understanding of slop failure mechanism, slope
movement, geology, geomorphology and hydrogeology is essential to carry out a landslide
susceptibility (Fell et al., 2008). In recent years, some physically-based models have been conducted to
study the mechanism of rainfall-induced landslides and infiltration analysis for individual slopes (e.g.
Lee et al., 2009; Cascini et al., 2010), and models generally combining an infinite stability model and a
hydrological model concerning topographic, geotechnical and hydrologic parameters for regional
assessment have been developed (e.g. Salciarini et al., 2006; Monstrasio et al., 2011; Kim et al., 2014).
Among the approaches, to facilitate the improvement of the landslide susceptibility the Geographic
Information Systems (GIS) are usually applied with its power to process spatial data (Carrara et al.,
1999; Dai et al., 2002b; Zhou et al., 2003; Neuhäuser et al., 2012). However, one of the drawbacks in
the physically-based approaches is prohibitive data requirements and therefore may be appropriate for
small areas (Dai et al., 2002b; Giannecchini et al., 2012).
As to Fengjie, existing studies revealed that rainfall is the main triggering factor for landslides, and
relevant researches were mainly about landslides distribution and slope failure mechanisms (e.g. Zhang
et al., 2004; Xu, 2005; Qi et al., 2006; Wang, 2007; Li, 2010; Yang et al., 2012), however, few landslide
inventory maps and landslide susceptibility maps were involved. Moreover, after the impounding of



Three Gorges Project in 2003, the environment in Fengjie has experienced large changes (Liu, 2005)
hence the landslides events in the past may not be a good indication to implement landslide assessment.
Thus, to carry out rainfall-induced landslide susceptibility map in Fengjie using the traditional methods
(e.g. Heuristic methods and statistical approaches) may not seem to be a good choice.

5       The objective of this paper is to carry out a quantitative assessment of landslide susceptibility in new
urban area of Fengjie county. The study develops an infinite stability model using Geo-studio software
concerning rainfall infiltration to obtain safety factor for individual slopes, then combining the
calculation results with artificial neural network (ANN) to figure out the relationship between
influencing factors and potential landslides, based on the trained model, using GIS, a landslide
susceptibility assessment map could be made.

## 2  The Study area

Fengjie County, passed through by the Yangtze River, lies in northeastern Chongqing Municipality. It is
in the Three Gorges area, which separates the Sichuan Basin and Jianghan Basin (Li et al., 2001). The
new urban area of Fengjie County, also named Sanmashan urban area, at present less than 7 km$^2$, is
distributed mainly along the north bank of Yangtze River (Fig. 1). On the north bank there are three
branches of Yangtze River, Caotang river (15km), Meixi river (40km) and Zhuyi river (20km), and
these rivers divide the north bank into three piece areas, Kouqianpian, Lianhuachi and Baotaping from
west to east. To settle thousands of immigrants because of Three Gorges Project, the new urban area
has experienced a rapid construction since 1996 (Fig. 2).

### 2.1  Geomorphological and geological settings

Fengjie County belongs to eastern Sichuan Basin, is located in the joint of Sichuan syneclise, fold
belt of Upper Yangtze platform and secondary structural belt of Daba platform, and folding
deformation was the dominant tectonic activity (Yang et al., 2012). The study area is situated in the
Three Gorges area which is characterized by continuous mountains, cliffs and deep river valleys.
Episodic intense tectonic movement and river incision during the quaternary are considered as the
primary formation reason of the Gorges (Li et al., 2001). The folds in the study area show a wide-slow
character and rock stratum trends in most parts of the area are in the direction E-W, approximately
paralleling the flow direction of Yangtze river with dip angles range from $5^o$ to $30^o$ (Luo et al., 2005).
The geological and structure features to a large extent influence the morphology. The layers cropping
out in the study area mainly belong to Triassic Jianglingjiang Formation and Badong Formation (Chang
et al., 2005). Composed of clastics and carbonate rocks, the Badong Formation almost distributes in the
whole area. The third number of Badong Formation ($T_2b^3$), composed of limestone, marlstone and
argillaceous limestone, dominating the stratum on the north bank, making the bank slopes with average
angles range from $25^o$ to $65^o$. In contrast, topography of the south bank is relatively slow ( Luo et al.,
2005; Xu, 2005; Yang et al., 2012). It is considered that the process of valleys incised by the Yangtze
River accounts for the deformation and fracture of stratum in some parts of the area, and the
well-developed cutting layered joints are a major feature on the north bank (Luo et al., 2005). The
weathering processes have significant influences on the properties of the widespread marlstone and
argillacous limestone, resulting in brittleness and fragility, thus the slopes in the area composed of
those rocks are prone to slide under certain triggering factors (Zhang, 2004; Chang et al., 2005).





## 2.2 Landslide occurrence and characteristics

Characterized by wet summers and autumns, with an annual mean precipitation ranges from 1126.7 mm to 1140.9 mm, Fengjie County is one of heavy rainfall centers in Three Gorges area. With complex geologic structure, fractured stratum and well developed gullies, the study area is a place prone to landslides and collapse (Ouyang et al., 2005). The landslides events in the study area are in accordance with the rainfall events (Zhang et al., 2005; Ma et al., 2009), heavy rainfall-induced landslides, continuing moderate-heavy rainfall-induced landslides are identified as the main types of landslides (Ma et al., 2005).

There are two main bank slopes, viz., rocky bank slopes and deposit bank slopes, of which the deformation and failure patterns were mainly classified as bending, cracking, cambered sliding, sliding-falling and flowing, etc. (Chang et al., 2004). It is observed that in the area the quantity of high soil bank slopes is larger than high rocky bank slopes, thus under rainfall shallow soil landslides or soft-bedrock landslides are more likely to be triggered. Fig.3 shows a shallow accumulative rainfall-induced landslide on the north bank in Fengjie. Loose accumulative landslides and bedrock landslides were reported as two main types in the area (Chen et al., 2005; Xu, 2005) and in the whole Three Gorges area the accumulative landslides took the largest proportion (Zhang and Liu, 2006).

# 3 Data and methods

In order to obtain rainfall-induced landslide susceptibility model, we developed an infinite stability model using Geo-studio software concerning rainfall infiltration to obtain safety factor for individual slopes, then combining the calculation results with ANNs to figure out the relationship between influencing factors and potential landslides, in that way the susceptibility model was achieved. Therefore, to carry out slope stability analyses is a pre-requisite. Considering the geological setting, the available data and the characteristics of the study area, an limit equilibrium method, namely Morgenstern-Price slice method was employed. Then, the safety factors were calculated via Geo-studio software (Slope/W module and Seep/W module) concerning rainfall infiltration into slopes.

Even the study area is a small region less than 7km$^2$, obtaining safety factors based on physical approaches for all potential slides is a huge task. In order to obtain sufficient slope stability calculation results for ANN performing the susceptibility model, according to detailed field survey by Wen et al., (2006), an analysis sample database (ASD) for the slope stability analyses was build by means of uniform design method, and slope stability influencing factors, e.g., rainfall, slope angle, slope height and cohesion were covered. On the basis of the ASD, safety factors were computed and then were utilized as training samples by ANN. In the last step, combining the trained ANN and survey data of the study area, the computation of safety factors under different rainfall were integrated in the GIS, and the rainfall-induced landslide susceptibility mapping in Fengjie could be made.

## 3.1 Data preparation

It is generally considered that landslides causing factors can be grouped into quasi-static factors and dynamic factors (Dai and Lee, 2001; Fall et al., 2006), for regional rainfall-induced landslide susceptibility assessment the quasi-static factors to be inputted are related to geomorphology, geology, land-use while for specific slopes slope geometry and geotechnical parameters should be considered (Lacasse and Nadim, 2011). The dynamic factors are usually rainfall and earthquake. To analyze the stability of a slope, the stability influencing factors, slope geometry and geotechnical parameters are




taken into consideration. Previous studies revealed that slope angle and slope height are critical
influencing factors in the study area (Zhou et al., 2004; Wen et al., 2011). Friction angle, cohesion and
weight in addition were taken to implement the stability analysis. The relationships between landslide
and rainfall and the prediction models in Chongqing region were studied by many researchers (e.g.
Chen et al., 2005; Zhang et al., 2005; Ma et al., 2009; Fan et al., 2012), daily rainfall or 24-hour rainfall
was normally taken as prediction index, however, cumulative rainfall based models failed to conduct
the prediction (Chen et al., 2015). Additionally, the seasonal fluctuation of reservoir level proved to be
related to the displacement for the colluvial landslides in the Three Gorges area (Du et al., 2013). With
regard to the study area, weights of influencing factors including the reservoir level were calculated
with analytic hierarchy process method by Wen et al. (2011) and the study indicated it is not particular
critical. Notwithstanding this, there is some uncertainty about the importance of reservoir level as few
studies coupling rainfall and reservoir level have been involved in related literature. Hence, in this
study the stability influencing factors selected were slope angle, slope height, cohesion, friction, weight
and rainfall (24-hour rainfall).
Based on the detailed field survey, the variations of influencing factor values in the study are
described in Table 1. The intensity of rainfall may vary in different elevations and terrains of an area
(Segoni et al., 2014), in order to make the physical based analysis feasible to do, the rainfall value
adopted here is in the form of mean daily rainfall.
To build the ASD, one of experiment design methods, uniform design method was adopted to make
the samples cover those 6 factors and guarantee sufficient experiment levels. The experiment levels of
the factors were designed as 30, accordingly, an uniform design table $U30*(30^{13})$ was utilized.
Afterwards, the MATLAB software was employed to divide the range value of each factor into 30
levels uniformly, and then combine the divided levels of the factors together. In this way, the ASD was
built (see Table 2).

**3.2    Computation of safety factors**
With regard to rainfall-induced landslide stability analysis, conceptual infiltration models (e.g.,
Green-Ampt model) combining with slope stability methods, analytical solutions and numerical
simulations are usually adopted. The use of conceptual infiltration models have been limited because
they usually simplify the infiltration problems (Ng and Shi, 1998), even analytical solutions have been
carried out by many researchers (e.g. Iverson, 2000; Chen et al., 2001; Rahardjo et al., 2005; Tsai and
Yang, 2006), the problems have not been satisfactorily addressed. With the character of high
non-linearity, the soil hydraulic properties were studied only by making assumptions when using
analytical solutions. The numerical analysis in conjunction with computer programs has an advantage
over analytical solutions because it could incorporate more advanced and sophisticated models to
analyze infiltration process in slopes under rainfall conditions (Zhang et al., 2011). Among them, the
commercial software Seep/W and Slop/W are often used for slope stability analysis under rainfall
condition. In addition, to obtain the safety factor for a slope, Bishop's simplified method (e.g. Rahardjo
et al., 2007; Wang et al., 2010) and Morgenstern-price method (e.g. Casagli et al., 2006; Cascini et al.,
2010) could be adopted. Once the ASD was completed, geometric models and hydraulic-mechanical
properties of the materials were determined. Hence, related parameters and coefficients for numerical
analysis were obtained. The calculated safety factors of 30 sample groups are described in Table 2.

**3.3    Data processing**




Similar to the brain, the artificial neural networks (ANNs) have great capability to learn from a set of
selection data with multiple computer algorithms. The ANNs have been proved to be useful in
modeling non linear and complex relationships between the input data and the desired output target.
Different types of neural networks have different learning rules and structures. The back propagation
(BP) algorithms trained neural network, also known as BP neural networks are commonly used with
the prediction performance of robustness and simplicity (Kurup and Dudani, 2002; Jang et al., 2004).
The applications of ANNs have been widely expanded into a variety of domains and the relevant
literature is too large. With regard to earth science, the ANNs have been adopted as important modeling
tools to conduct landslide susceptibility (Ermini et al., 2005; Melchiorre et al., 2008; Kawabata and
Bandibas, 2009).
In this study, the BP ANN with a single hidden layer was employed (Fig. 4). As illustrated in the
schematic, three layers (the input layer, the hidden layer and the output layer) and their
interconnections constitute the neural network. In the network, each layer is a group of several neurons,
$x_j$ denotes input variable to the neuron $j$, $w_{ij}$ connection is the weight from neuron $j$ in the input layer
to neuron $i$ in hidden layer and $o_k$ is the output of neuron $k$. Besides, $\phi$ in the hidden layer and $\psi$ in the
output layer are activation functions respectively, $\theta_i$ and $a_k$ stand for threshold weights of the neurons.
It is important to choose appropriate quantity of neurons in hidden layer, hence the trail and error
method was used in this paper, and the number of neurons was determined as 7. In addition to test the
prediction performance of the ANN, the mean squared error (MSE) is often adopted as the performance
index. The training algotithms, the structure, together with training samples govern the final
performance of the network.
Prior to the training process, using the software Matlab, the ASD has to be normalized on nominal
scales as binary numbers in case of convergence problems (see Table 3). In the training process (Fig. 5),
the input variables in the training sample were slope angle, slope height, cohesion, friction, weight and
rainfall, and the output is safety factors. As can be seen form Fig. 6 and Fig. 7, the performance error
gradually decreases to 1E-5, which means convergence of the training was good, moreover,
performance of the trained networks are satisfactory in terms of logical reasoning and internal
relationship. Then the trained ANN is applied to model the influencing factors derived from detailed
field survey, finally the outputs would be mapped within a GIS.

### 3.4  Thematic data layers

As mentioned above, landslide influencing factors could be divided into quasi-static factors and
dynamic factors. In terms of the 6 influencing factors, rainfall is regarded as dynamic factor while the
others are quasi-static factors, classification and category of landslide influencing factors are shown in
Table 4. Based on precipitation amount, in the study area the mean daily rainfall are classified in 5
types (Zhang et al., 2005; Fan et al., 2012).
According to the field survey results, 5 thematic data layers pertaining to the quasi-static factors,
slope angle, slope height, cohesion, friction and weight were made within a GIS. Each influencing
factor was first built as vector layer then converted to raster layer, and the data in each raster layer was
classified in order to facility the management and computation. The 5 thematic layers are listed in Fig.
41 8.

## 4  Rainfall-induced landslide susceptibility assessment



### 4.1 Landslide susceptibility assessment

The thematic data layers were integrated combined with the trained ANN, in this way rainfall-induced landslide susceptibility assessment was achieved under 5 types of rainfall events. Then the assessment results were mapped for different rainfall events within the GIS, we choose a typical value for each rainfall type, and the calculations values of different rainfall are shown in Table 5. With regard to the assessments, they were classified into 5 grades according to the computation values of slope instability (Table 6). This classification was conducted by means of natural break points method. It is found that under light rainfall, moderate rainfall and even heavy rainfall VH and H zones could hardly be discerned, which means landslides in the area may not be induced by the rainfall conditions when its mean daily rainfall less than 50 mm. Fig. 9 shows the landslide susceptibility assessment under rainstorm and severe rainstorm in the study area. Furthermore, statistical results about the landslide susceptibility zonation under rainstorm and severe rainstorm were calculated concerning these frequent weathers in the study area (Table 7).

As can be observed from Fig. 9, It is also found that M zones, H zones and VH zones are mainly distributed along the Yangtze River and its three branches, which indeed should be the case since a large number of landslides have been observed in those locations. The bank slopes with fractured stratum, weak rocks and deposits may provide a basis for landslide occurrence under rainstorm and severe rainstorm. As shown in Table 7, it is observed that the H zones distributed in the study area under rainstorm account for a small portion (4.45%) while it is not the case under severe rainstorm (24.80%). The contrast of the landslide susceptibility zonation under rainstorm and severe rainstorm reveals the important role played by the process of the rainstorm developing to severe rainstorm, which could be verified by the landslides occurrence reported under severe rainstorms in recent years. Therefore, care should be taken to the forecast of rainstorm and severe rainstorm as the total percent areas of H zones and VH zones under severe rainstorm occupy more than a quarter (27.69%) in the study area. Another big variation is VL zones under rainstorm ( 27.9 %) and severe rainstorm (11.8 %), the decrease (i.e. 16.10%) of the stable area maybe account for the initial infiltration process under severe rainstorm.

### 4.2 Validation

To validate the landslide susceptibility assessment, we made a thorough investigation about landslides in the area from 1998 to 2014. With regard to the triggering factor, rainfall, mostly rainstorm (mean daily rainfall over 100mm), accounting for the overwhelming majority. 58 rainfall (rainstorm)-induced landslides are presented in Table 8 and Fig. 10, locations, occurrence time and general directions of these landslides are clearly identified. All of the landslides were directly triggered by rainfall, however, among these landslides, some occurred under the combination of rainfall and river erosion, which needs more efforts to clarify the major cause.

As can be seen from Fig. 9 and Fig. 10, the actual landslides are accordance with the assessment results, locations of these actual landslides mostly fall on H zones and VH zones in Fig. 9. However, unexpected results are found about some landslides, namely L-10, L-13, L-19, L-20, L-26, L-37, L-43, L-47 and L-53, which accounts for 15.5 % of the whole landslides.

Notwithstanding the apparent satisfactory results, the susceptibility assessment could not be proved robust, as the database chosen for training ANNs and performing susceptibility were from the investigation completed before 2006. What's more, after the impounding of Three Gorges Project in 2003, the environment in Fengjie has experienced large changes, thus the bank slopes would have a fair



chance to slide owing to water-level rising. As shown in Table 8, landslides experience a sharp increase
in 2003. It would be more convincing to use landslide data after 2006 to verify feasibility of the
susceptibility assessment model. Hence we choose L-1, L-2, L-3, L-4, L-5, L-6, L-10, L-11, L-13, L-24,
L-25, L-27 and L-45 from the whole actual landslides, and a good validation was achieved except L-10
and L-13.

## 5  Discussion

Generally speaking, qualitative methods are effective to carry out an rainfall-induced landslide
susceptibility on condition that there are enough historical data. However, the complete and unbiased
database of rainfall intensity and duration, landslide magnitude and volume, slope failure patterns and
landslide processes are not available in the study area. In this regard, a quantitative method based on
detailed investigation seems to be a better option. Stability analyze of rainfall-induced landslides using
Geo-studio software is a basis for the susceptibility model, and good performance of the model may be
attributed to the fact that soil landslides account for the majority of the chosen landslides as the
software proved to be excellent to analyze rainfall-induced soil slope failure. When it comes to rock
slide and more complex landslide type, stability analysis and the ASD to train ANNs should be more
rigorous.
The assessment results in this study are also in accordance with the relationship between rainfall and
landslides in some relevant literature (Xu, 2005; Zhang et al., 2005), where the studies were conducted
by statistical methods. On the whole, it is suggested that under rainstorm and severe rainstorm
conditions care should be taken to notice the landslide development. With regard to the susceptibility
assessment results, the major finding is that the high susceptibility zones are mainly distributed along
the Yangtze River and its three branches. That bank slopes consisting of fractured stratum, weak rocks
and deposits may be considered as a major reason. Owing to river erosion and rainfall infiltration, bank
slopes may have a higher chance to slide. A rainfall-induced landslide, Baiyian landslide (L-44, as
shown in Fig. 10), occurred in 22 July 2003 (Zhang et al., 2004) was a representative as the
conjunction result of rainfall, water-level rising and geological conditions. However, our work did not
take the water-level rising effect into consideration as there exist problems to quantify the effect. Hence
it should be more careful when applied the rainfall-induced landslide susceptibility model in the study
area.
Due to the uncertainty lies in rainfall patterns and slope properties, it is difficult to precisely predict a
landslide, and the slope failures may not in accordance with the predictions. In this study, the effect of
mitigation measures in addition were not taken into account as the mitigation and rehabilitation
measures have been adopted to against the landslides since the construction of the new urban area (Xu,
2005). Moreover, it is complicated to incorporate the uncertainties of geotechnical data. Hence, in
terms of the quantitative methods, the combination of reliability and physically-based analysis may be
promising to address the problems.
Although the quantitative method could reasonably delineate landslide susceptibility in the study
area, the knowledge about rainfall-induced landslide occurrence, influencing factors and the infiltration
processes are still limited.

## 6  Conclusion

In this work, a quantitative assessment of rainfall-induced landslide susceptibility in new urban area of
Fengjie County was carried out. The methodology presented in this paper was based on the





combination of mechanical stability analysis and ANN and of GIS and detailed field investigation. The detailed field survey could provide valuable geomorphological, geological and geotechnical information about the study area, which is a basis for landslide susceptibility assessment, in particular, for the physically based landslide susceptibility assessment presented in this paper. Based on the field survey and relevant literature, information about the geological and geotechnical parameters for slope stability analysis via numerical simulation software Geostudio (Slope/W module and Seep/W module) was concerning slope angle, slope height, cohesion, friction, weight and rainfall. Then, the safety factors of site -specific slopes from the ASD were calculated. The employment of ANN was a bridge between the individual slope stability analysis and the overall slope stability analyses in a regional scale, and within a GIS the quantitative assessment of the landslide susceptibility was mapped. Subsequently, 58 actual rainfall (rainstorm)-induced landslides occurred in the study area from 1998 to 2014 were used to verify the susceptibility assessment, and satisfactory results were obtained.

The landslide susceptibility zonation implies that slopes in more than a quarter of the study area are prone to landslides under rainstorm and severe rainstorm while the overall landslide susceptibility under light rainfall, moderate rainfall and even heavy rainfall are rather low. In spite of the costly countermeasures, the problem of landslides still poses a threat owing to the increasing rainstorm events in the area (Lin and Yang, 2014), the new urban area of Fengjie County fails to be a good place to hold tens of thousands of the residents. As a result, a new place, the West District of Fengjie (in the west of the study area) has been planned and constructed to be another urban area since 2010.

## Acknowledgments

The field work was carried out in collaboration with the Institute of Exploration Technology, China Geological Survey. The authors are grateful to Zhongsheng Xie and Shengwei Shi for their assistance.

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



2     Fig. 3. A large shallow accumulative rainfall-induced landslide beside the Yangtze River.

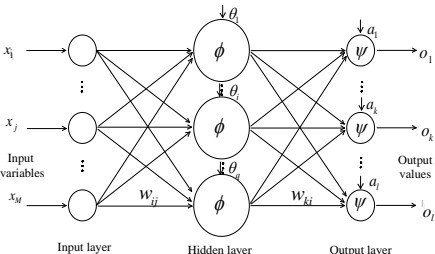

5     Fig. 4. A three layered BP Artificial Neural Network

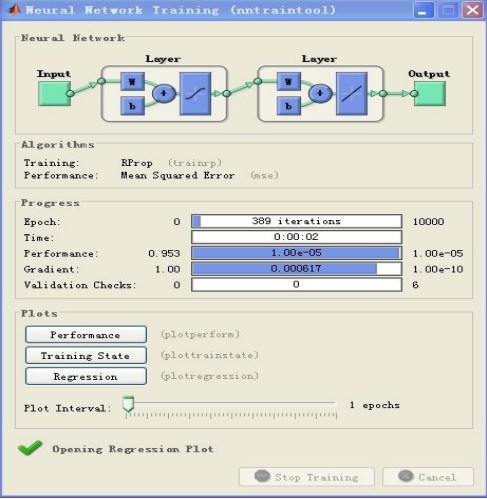

8     Fig. 5.The neural network training process




2 Fig. 6. The neural network training sample performance graph

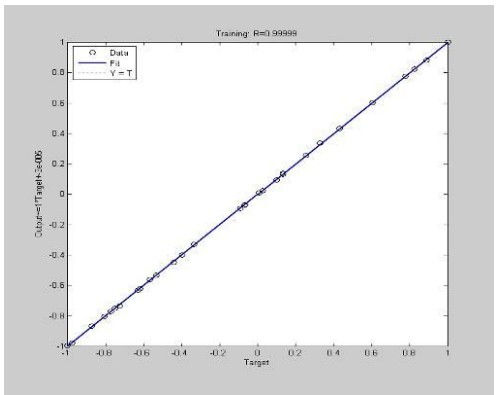

5 Fig. 7. The regression of Neural Networks

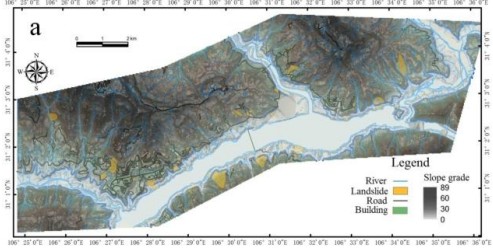



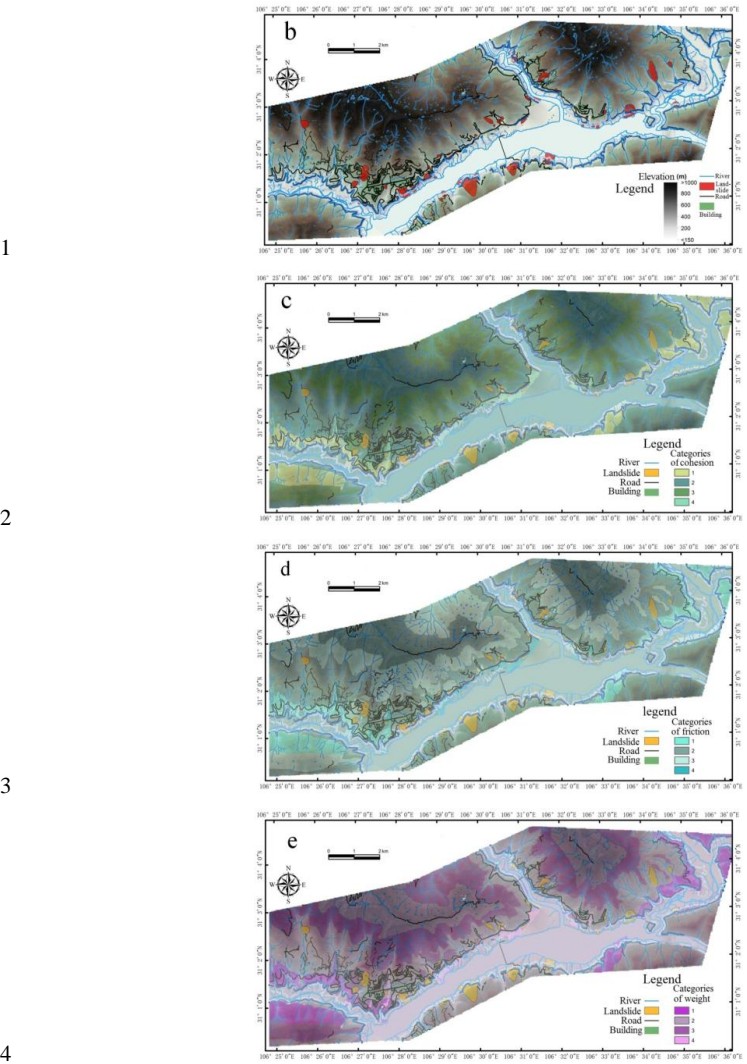

5   Fig. 8. Thematic layers of 5 influencing factors in new urban area of Fengjie County based on detailed field

6         survey : (a) slope grade, (b) slope height, (c) cohesion, (d) friction, (e) weight





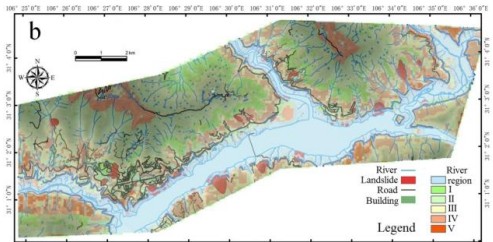

2             Fig. 9. Landslide susceptibility mapping in new urban area of Fengjie County under heavy rainfall events:

3                     (a) Rainstorm, (b) Severe rainstor




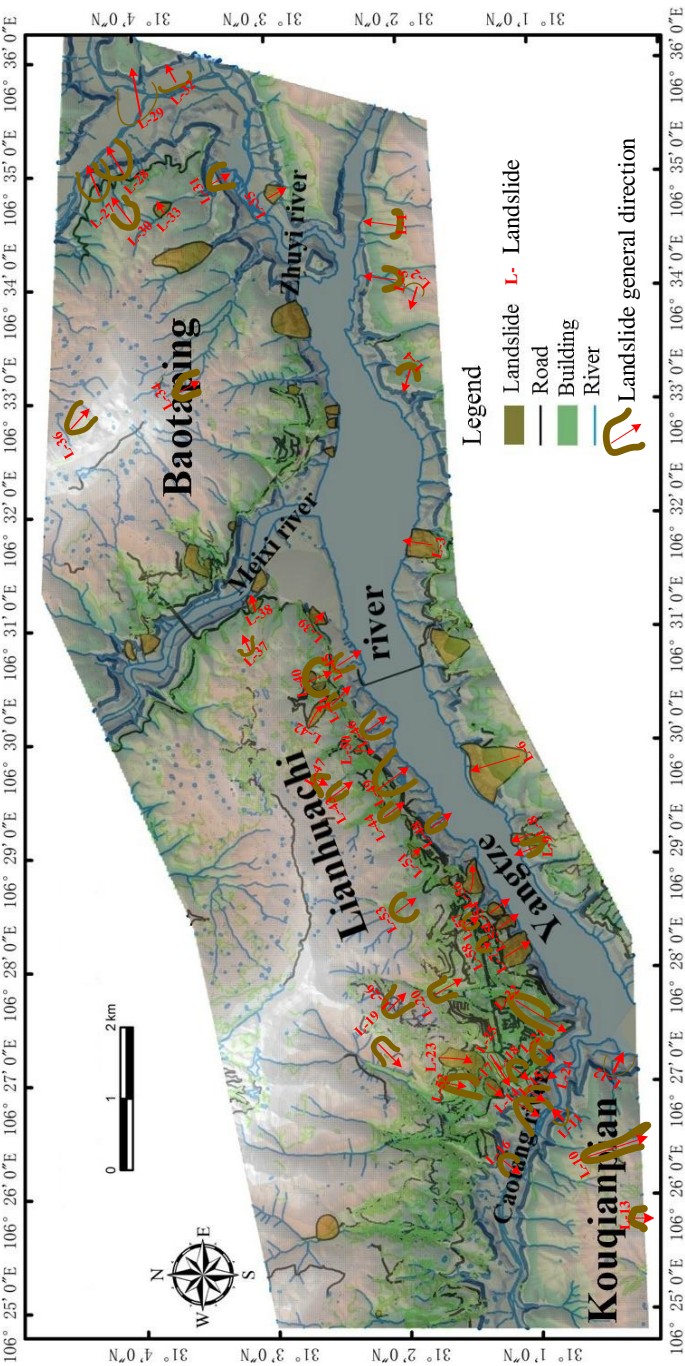

Fig. 10. Detailed investigation of rainfall-induced landslides in urban area of Fengjie County (from 1998 to 2014)





Table 1

Variation of influencing factor values in the study area

| Influencing factors | Slope angle | Slope height | Cohesion | Friction | Weight | Rainfall |
|---|---|---|---|---|---|---|
| Unit | $\alpha$ (°) | $h$ (m) | $c$ (MPa) | $\varphi$ (°) | $\gamma$ (kN/m$^3$) | $p$ (mm/d) |
| Ranges | 25~65 | 8~14.9 | 0.5~100 | 4~42 | 13.7~19.6 | 0~121.4 |

Table 2

Analysis sample database of influencing factors and safety factors

| Sample groups | Slope angle $\alpha$ (°) | Slope height $h$ (m) | Cohesion $c$ (MPa) | Friction $\varphi$ (°) | Weight $\gamma$ (kN/m$^3$) | Rainfall $p$ (mm/d) | Safety factors |
|---|---|---|---|---|---|---|---|
| Group 1 | 25.00 | 8.71 | 31.38 | 21.03 | 17.16 | 100.47 | 2.67 |
| Group 2 | 26.38 | 9.67 | 65.69 | 39.38 | 14.51 | 75.35 | 3.28 |
| Group 3 | 27.76 | 10.62 | 100.00 | 17.10 | 18.18 | 50.23 | 4.20 |
| Group 4 | 29.14 | 11.57 | 27.95 | 35.45 | 15.53 | 25.12 | 2.68 |
| Group 5 | 30.52 | 12.52 | 62.26 | 13.17 | 19.19 | 0.00 | 2.27 |
| Group 6 | 31.90 | 13.47 | 96.57 | 31.52 | 16.55 | 104.66 | 4.43 |
| Group 7 | 33.28 | 14.42 | 24.52 | 9.24 | 13.90 | 79.54 | 1.16 |
| Group 8 | 34.66 | 8.00 | 58.83 | 27.59 | 17.57 | 54.42 | 2.92 |
| Group 9 | 36.03 | 8.95 | 93.14 | 5.31 | 14.92 | 29.30 | 4.07 |
| Group 10 | 37.41 | 9.90 | 21.09 | 23.66 | 18.58 | 4.19 | 1.60 |
| Group 11 | 38.79 | 10.86 | 55.40 | 42.00 | 15.94 | 108.84 | 3.98 |
| Group 12 | 40.17 | 11.81 | 89.71 | 19.72 | 19.60 | 83.72 | 3.07 |
| Group 13 | 41.55 | 12.76 | 17.66 | 38.07 | 16.96 | 58.61 | 0.84 |
| Group 14 | 42.93 | 13.71 | 51.97 | 15.79 | 14.31 | 33.49 | 1.51 |
| Group 15 | 44.31 | 14.66 | 86.28 | 34.14 | 17.97 | 8.37 | 0.38 |
| Group 16 | 45.69 | 8.24 | 14.22 | 11.86 | 15.33 | 113.03 | 0.78 |
| Group 17 | 47.07 | 9.19 | 48.53 | 30.21 | 18.99 | 87.91 | 2.61 |
| Group 18 | 48.45 | 10.14 | 82.84 | 7.93 | 16.34 | 62.79 | 1.73 |
| Group 19 | 49.83 | 11.09 | 10.79 | 26.28 | 13.70 | 37.68 | 1.13 |
| Group 20 | 51.21 | 12.04 | 45.10 | 4.00 | 17.36 | 12.56 | 1.26 |
| Group 21 | 52.59 | 13.00 | 79.41 | 22.34 | 14.72 | 117.21 | 2.42 |
| Group 22 | 53.97 | 13.95 | 7.36 | 40.69 | 18.38 | 92.10 | 0.64 |
| Group 23 | 55.34 | 14.90 | 41.67 | 18.41 | 15.73 | 66.98 | 0.88 |
| Group 24 | 56.72 | 8.48 | 75.98 | 36.76 | 19.40 | 41.86 | 3.63 |
| Group 25 | 58.10 | 9.43 | 3.93 | 14.48 | 16.75 | 16.74 | 0.43 |
| Group 26 | 59.48 | 10.38 | 38.24 | 32.83 | 14.11 | 121.40 | 2.22 |
| Group 27 | 60.86 | 11.33 | 72.55 | 10.55 | 17.77 | 96.28 | 2.26 |
| Group 28 | 62.24 | 12.28 | 0.50 | 28.9 | 15.12 | 71.17 | 1.33 |
| Group 29 | 63.62 | 13.23 | 34.81 | 7.36 | 18.79 | 46.05 | 0.94 |
| Group 30 | 65.00 | 14.19 | 69.12 | 24.97 | 16.14 | 20.93 | 2.46 |



Table 3

The Normalized database

| Sample groups | Slope angle | Slope height | Cohesion | Friction | Weight | Rainfall | Safety factors |
|---|---|---|---|---|---|---|---|
| Group 1 | -1 | -0.7931 | -0.37931 | -0.44774 | 0.172414 | 0.655172 | 0.131768 |
| Group 2 | -0.93103 | -0.51724 | 0.310345 | 0.14701 | -0.72414 | 0.241379 | 0.430408 |
| Group 3 | -0.86207 | -0.24138 | 1 | -0.57518 | 0.517241 | -0.17241 | 0.886279 |
| Group 4 | -0.7931 | 0.034483 | -0.44828 | 0.019564 | -0.37931 | -0.58621 | 0.133745 |
| Group 5 | -0.72414 | 0.310345 | 0.241379 | -0.70263 | 0.862069 | -1 | -0.0665 |
| Group 6 | -0.65517 | 0.586207 | 0.931034 | -0.10788 | -0.03448 | 0.724138 | 1 |
| Group 7 | -0.58621 | 0.862069 | -0.51724 | -0.83007 | -0.93103 | 0.310345 | -0.6178 |
| Group 8 | -0.51724 | -1 | 0.172414 | -0.23533 | 0.310345 | -0.10345 | 0.255871 |
| Group 9 | -0.44828 | -0.72414 | 0.862069 | -0.95752 | -0.58621 | -0.51724 | 0.823486 |
| Group 10 | -0.37931 | -0.44828 | -0.58621 | -0.36277 | 0.655172 | -0.93103 | -0.39827 |
| Group 11 | -0.31034 | -0.17241 | 0.103448 | 0.231973 | -0.24138 | 0.793103 | 0.778986 |
| Group 12 | -0.24138 | 0.103448 | 0.793103 | -0.49022 | 1 | 0.37931 | 0.329543 |
| Group 13 | -0.17241 | 0.37931 | -0.65517 | 0.104528 | 0.103448 | -0.03448 | -0.77206 |
| Group 14 | -0.10345 | 0.655172 | 0.034483 | -0.61766 | -0.7931 | -0.44828 | -0.44326 |
| Group 15 | -0.03448 | 0.931034 | 0.724138 | -0.02292 | 0.448276 | -0.86207 | -1 |
| Group 16 | 0.034483 | -0.93103 | -0.72414 | -0.74511 | -0.44828 | 0.862069 | -0.80519 |
| Group 17 | 0.103448 | -0.65517 | -0.03448 | -0.15036 | 0.793103 | 0.448276 | 0.100124 |
| Group 18 | 0.172414 | -0.37931 | 0.655172 | -0.87255 | -0.10345 | 0.034483 | -0.3335 |
| Group 19 | 0.241379 | -0.10345 | -0.7931 | -0.27781 | -1 | -0.37931 | -0.63066 |
| Group 20 | 0.310345 | 0.172414 | -0.10345 | -1 | 0.241379 | -0.7931 | -0.56588 |
| Group 21 | 0.37931 | 0.448276 | 0.586207 | -0.40525 | -0.65517 | 0.931034 | 0.005686 |
| Group 22 | 0.448276 | 0.724138 | -0.86207 | 0.189491 | 0.586207 | 0.517241 | -0.87244 |
| Group 23 | 0.517241 | 1 | -0.17241 | -0.5327 | -0.31034 | 0.103448 | -0.75179 |
| Group 24 | 0.586207 | -0.86207 | 0.517241 | 0.062046 | 0.931034 | -0.31034 | 0.603461 |
| Group 25 | 0.655172 | -0.58621 | -0.93103 | -0.66015 | 0.034483 | -0.72414 | -0.97726 |
| Group 26 | 0.724138 | -0.31034 | -0.24138 | -0.0654 | -0.86207 | 1 | -0.09221 |
| Group 27 | 0.793103 | -0.03448 | 0.448276 | -0.78759 | 0.37931 | 0.586207 | -0.07145 |
| Group 28 | 0.862069 | 0.241379 | -1 | 1 | -0.51724 | 0.172414 | -0.53325 |
| Group 29 | 0.931034 | 0.517241 | -0.31034 | -0.891 | 0.724138 | -0.24138 | -0.72658 |
| Group 30 | 1 | 0.793103 | 0.37931 | -0.32029 | -0.17241 | -0.65517 | 0.026452 |





Table 4

Classification and category of landslide influencing factors

| Landslide influence factors | | | | | | Landslide risk |
|---|---|---|---|---|---|---|
| Rainfall | Slope angle(o) | Slope height | Cohesion | Friction | Weight | |
| (0, 10)mm | (0, 15)/ [60, 90) | [100, ∞) m | I | I | I | Very low |
| [10, 25) mm | [15, 25)/[45, 60) | [50,100)m | II | II | II | Low |
| [25, 50) mm | — | [25, 50)m | — | — | — | Moderate |
| [50, 100) mm | [25, 35) mm | [15, 25)m | III | III | III | High |
| [100, ∞) mm | [35, 60) mm | [8, 15)m | IV | IV | IV | Very high |

In the table, category of cohesion IV, III, II, I denote the value of cohesion vary from 0.5 to 20 MPa, from 20 to 50 MPa, from 50 to 80 MPa, and 80 to 100 MPa, respectively; category of friction IV, III, II, I denote the value of friction angle vary from 4 °to 10 °, from 10 °to 25 °, from 25 °to 35 °, and 35 °to 42 °, respectively; in terms of weight, the category I, II, III, IV denote the value of weight vary from 13.7 to 15.2 kN/m3, from15.2 to 16.8 kN/m3, from16.8 to 17.6 kN/m3, and 17.6 to 19.6 kN/m3, respectively.

Table 5

Calculations value of different rainfall

| Rainfall types | Range | Calculations value |
|---|---|---|
| Light rainfall | (0, 10) mm | 5 |
| Moderate rainfall | [10, 25) mm | 15 |
| Heavy rainfall | [25, 50) mm | 37 |
| Rainstorm | [50, 100) mm | 75 |
| Severe rainstorm | [100, ∞) mm | 121 |

Table 6

Classification of landslide susceptibility assessment

| Classifications | Landslide susceptibility description |
|---|---|
| I | Very low(VL) |
| II | Low (L) |
| III | Moderate (M) |
| IV | High (H) |
| V | Very high (VH) |

Table 7

landslide susceptibility zonation under rainstorm and severe rainstorm

| Landslide susceptibility zonation | Rainfall conditions | | Variation |
|---|---|---|---|
| | Rainstorm (Percent area %) | Severe rainstorm (percent area %) | |
| VL Zones | 27.90% | 11.80% | -16.10% |
| L Zones | 45.20% | 40.19% | -5.02% |
| M Zones | 22.45% | 20.32% | -2.13% |
| H Zones | 4.45% | 24.80% | 20.35% |
| VH Zones | 0.00% | 2.89% | 2.89% |

The area of total zones is the study area



Table 8

Location and occurrence time of rainfall-induced landslides in urban area of Fengjie County (from 1998 to 2014)

| Landslide ID | Location(name) | Time | Landslide ID | Location(name) | Time |
|---|---|---|---|---|---|
| L-1 | Kuimen | 2006.6 | L-30 | Miaowanzi | 2004.9 |
| L-2 | Daoziping | 2008.9 | L-31 | Yueliangping | 2002.5 |
| L-3 | Yujiafen | 2007.4 | L-32 | Caotanghe | 2000.7 |
| L-4 | Yanmenzi | 2006.6 | L-33 | Shangenbao | 2004.8 |
| L-5 | Kunniushi | 2007.6 | L-34 | Ziyang-4-she | 2001.3 |
| L-6 | Miaobao | 2007.6 | L-35 | Xiaooujiabao | 2004.5 |
| L-7 | Lanshiyao | 2000.10 | L-36 | Lengjiawan | 2008.4 |
| L-8 | Zicantuo | 2000.8 | L-37 | Yangjiawuchang | 2000.7 |
| L-9 | Minjiabao | 2000.8 | L-38 | Hongyadong | 1998.7 |
| L-10 | Shangbolin | 2007.7 | L-39 | Fj-Middle school | 2005.4 |
| L-11 | Xiabolin | 2006.4 | L-40 | Chatupo | 2000.6 |
| L-12 | Hualianshu | 2004.7 | L-41 | Tudiliang | 2000.8 |
| L-13 | Zhongzui | 2008.5 | L-42 | Chenjiawan | 2003.6 |
| L-14 | Shaojiabao | 2000.7 | L-43 | Jigongliang | 2003.7 |
| L-15 | Guojiabao | 2001.7 | L-44 | Baiyian | 2003.7 |
| L-16 | Kuangjiagou | 2000.7 | L-45 | Happy-zhongxue | 2007.6 |
| L-17 | Wangjiaping | 1998.7 | L-46 | Zhoujiawan | 2003.7 |
| L-18 | Dikuangju | 1998.7 | L-47 | Wangjiawan | 2003.8 |
| L-19 | Lijiagou | 1998.7 | L-48 | Yaoping | 2000.8 |
| L-20 | Gufang | 2003.7 | L-49 | Zhuanchang | 2000.8 |
| L-21 | Zhuyaozi | 2003.7 | L-50 | Yanjiapo | 2001.7 |
| L-22 | Liujiabao | 1998.7 | L-51 | Erpingzi | 2002.5 |
| L-23 | Shijialiang | 1998.7 | L-52 | Zhangjiawuchang | 2003.6 |
| L-24 | Toudaohe | 2009.7 | L-53 | Dahegou | 2000.7 |
| L-25 | Jixiegongsi | 2009.7 | L-54 | Dengzhanwo | 2003.7 |
| L-26 | Baiyangping | 2005.7 | L-55 | Yinliping | 2001.6 |
| L-27 | Chenjiawan | 2014.9 | L-56 | Laofangzi | 2001.6 |
| L-28 | Oujiabao | 2005.9 | L-57 | Sichouchang | 2002.6 |
| L-29 | Luojiawan | 2003.6 | L-58 | Houzishi | 2003.7 |