# Peer review of "Quantitative assessment of rainfall-induced landslide susceptibility in new urban area of Fengjie County, Three"

_Natural Hazards and Earth System Sciences, 2017_

## Referee Comment (RC1) · Anonymous Referee #1 · 10 May 2017

Comments about the manuscript 'Quantitative assessment of rainfall-induced landslide susceptibility in new urban area of Fengjie County, Three Gorges area, China' :

General comments:

The paper uses large quantity of data to assess regional bank landslide susceptibility in Fengjie town of Three-Gorges Reservoir area. Authors made great efforts trying to analyze, re-group different kinds of parameters which are necessary for a quantitative analysis. Meanwhile, validation of predicted results is conducted according to thorough field landslide investigation and recorded events of landslides in the past years. Maps are beautiful. Data are rich. All efforts from the authors are encouraged. But unfortunately, the paper failed to some major issues.

1. What is the author's real point of view on the influencing factor of reservoir water level? The title of the paper is focused on the analysis of rainfall-induced landslides, but for regional susceptibility mapping, the paper ignored the influencing factor of reservoir water level fluctuation only based on the simple explanation in Page 5 line 10-15. In fact, the authors hold an opposite point of view from page 7 lines 35-36, page 8 lines 1-2, which demonstrate the action of reservoir water level. So, what is the real author's recognition on the factor of reservoir water fluctuation? It seems to be a contradiction.

2. Data source and processing are not adequate. For quantitative assessment of landslide susceptibility, data is essential even the data source is always very difficult in a regional scale. Nevertheless, it is not tolerable in the paper that there are some unreasonable parameters considering the real soil/rock properties which significantly affect the reliability of calculation. Questions to table 1. (1) h, slope height, ranges 8-14.9? Normally, a natural slope is much higher as hundreds meters. But, in table 4 the selected range of h is higher than 100m. (2) c, cohesion, is presented in MPa which generally means a contact rock material not a soil? How this high cohesion is in relation to the shallow soil slope failures? (3)ÏŠ, weight, minimum is 13.7KN/m3, what kind of soil is it? The reviewer thinks the proper data source and processing are the first issue to ensure the reliability of the analysis.

3. Quantitative calculation of FS in ASD seems not to be realistic. Questions on FS in table 2. From the reviewer's experience in FS calculation of safety factor, FS = 0.38 of group 15 slope is calculated with parameters of friction angle 34.14, cohesion 86.28 Mpa, and 8.37 mm/day of rainfall. Is it corrected? But FS = 4.07 of group 9 slope is calculated with parameters of friction angle 5.31, cohesion 93.14 Mpa, and 29.30 mm/day of rainfall? These two FS values made me confused. Comparing to all FS values in Table 2, how to evaluate the reliability of FS? The reviewer thinks the training samples are the core for later susceptibility analysis. Please re-check your data source and your calculation.

Specific questions:

4. Are the key words present the key points of the paper? It should be re-considered?
5. Page 2 lines 43-44, "however, few landslides inventory maps...". In fact, very detailed landslide inventories, geological exploration, even many slope stabilization engineering have been already completed which is organized by central government. 6. Page 3 line 2, "hence the landslides events in the past may not be a good indication to implement landslide assessment". This statement is not geologically correct. The past is a good indication for future, especially for regional susceptibility mapping. We always use the inventory maps and recorded events as the data source which demonstrated the importance of the past. 7. Page 4 line 42, "the dynamic factors are usually rainfall and earthquake". Reservoir water level variation is certainly an important dynamic factor which affects the bank slope in this paper. Reservoir water level plays two key roles to destabilize the bank slope: soften soil/rock properties and changing the pore water pressure. Why the paper does not use this parameter? 8. Page 5 line 38 – 40, "Bishop's simplified method and Morgenstern-price method..... adopted". Which method is employed in the calculation?

For above comments, the reviewer think the paper needs to be greatly improved in three key issues and some specific points in order to reach a reliable regional landslide susceptibility map. So the reviewer would like to give the paper a rejection.

Please also note the supplement to this comment:
http://www.nat-hazards-earth-syst-sci-discuss.net/nhess-2017-99/nhess-2017-99-RC1-supplement.pdf
* * *

---

## Referee Comment (RC2) · Anonymous Referee #2 · 22 Jun 2017

The paper deals with an interesting topic, quantitative assessment of rainfall-induced landslide susceptibility at regional scale. Yet the paper has many major flaws and therefore it should be, in my opinion, rejected.

Major comments

The data and the results are not presented in a clear and well-structured way.

The method proposed by the Authors is not clear. In the Abstract they state the methodology is based on a "combination of mechanical stability analysis and artificial neural network (ANN) and of Geographic Information Systems (GIS ) and detailed field investigation". At the end of the introduction they state "The study develops an infinite

stability model using Geo-studio software concerning rainfall infiltration to obtain safety factor for individual slopes, then combining the calculation results with artificial neural network (ANN) to figure out the relationship between influencing factors and potential landslides, based on the trained model, using GIS, a landslide susceptibility assessment map could be made". Yet they do not clearly explain, in chapter 3, how the various parts of the procedure (i.e. data from field investigation, stability analyses, ANN, susceptibility mapping) work and interact. The hypotheses of the various analyses performed are not stated.

Concerning the data from the field investigation, how are the ranges of values presented in Table 1 retrieved? How did the Authors derive the 30 combinations of values reported in Table 2? Where do the original data come from? Are they representative of the field conditions in the whole study area or only in the areas where landslides occurred? For instance, the unit weight and the strength properties (extremely large ranges are reported for the cohesion and friction angle values) are representative of what type of soil/rock? How may landslides are there in the study area and in what type of soil/rock do they occur?

Concerning the stability analyses, are they conducted considering an infinite slope schematization (as written in page 3, line 6 and page 4, lines 19-20) or a boundary value problem in 2D (as one may infer from the fact they use the Geo-studio software)? In the latter case, what's the geometry of the considered slope(s)? The Authors do not provide any detail on the schematization of the slopes and on the shape and position of the sliding surfaces. What's the meaning of the 30 factors of safety computed (Table 2) in relation to the slopes and landslides present in the study area?

Concerning the ANN and susceptibility mapping, is the data from Table 3 (derived from Table 2) the only data used to train the ANN looking for a relationship between the six so-called "influencing factors" and the safety factor? If so, what's the significance of the trained ANN in relation to the slopes and landslides present in the study area? How were the FS values computed with the trained ANN converted into the five reported

classes (Table 6 does not report any range of values for FS)? What are the landslide data used for validation, years 1998-2014 or 2006-2014? In the paragraph starting at the end of page 7 the Authors state "after the impounding of Three Gorges Project in 2003, the environment in Fengjie has experienced large changes, thus the bank slopes would have a fair chance to slide owing to water-level rising .. It would be more convincing to use landslide data after 2006 to verify feasibility of the susceptibility assessment model." Does it mean the trained ANN is valid only for conditions in the study area successive to the impounding of Three Gorges Project in 2003? It appears that the Authors believe there's a role played by the reservoir level. If so, it should have been considered in the analyses.

Some of the sentences in the discussion session seem to disprove the relevance of the obtained results, e.g. "Our work did not take the water-level rising effect into consideration as there exist problems to quantify the effect. Hence it should be more careful when applied the rainfall-induced landslide susceptibility model in the study area. .. Due to the uncertainty lies in rainfall patterns and slope properties, it is difficult to precisely predict a landslide, and the slope failures may not in accordance with the predictions."

The use of the English language is not adequate.

---

## Author Comment (AC1) · 28 Jul 2017

Author comments Ref: nhess-2017-99

We appreciate the time taken to review this paper and constructive comments were given. Those comments are all valuable and helpful for revising and improving our paper, as well as the important guiding significance to our researches. We have studied comments carefully and have made correction as much as possible. The main corrections in the paper and the responds to the reviewer's comments are as follows:

Responds to the reviewer's comments: Reviewer#1:

1. What is the author's real point of view on the influencing factor of reservoir water level? The title of the paper is focused on the analysis of rainfall-induced landslides, but for regional susceptibility mapping, the paper ignored the influencing factor of reservoir water level fluctuation only based on the simple explanation in Page 5 line 10-15. In fact, the authors hold an opposite point of view from page 7 lines 35-36, page 8 lines 1-2, which demonstrate the action of reservoir water level. So, what is the real author's recognition on the factor of reservoir water fluctuation? It seems to be a contradiction.

We focus on the analysis of rainfall-induced landslide, however, we do not deny the influence of reservoir water level fluctuation on triggering landslide as can be seen from the landslide events in our paper. With regard to the influencing factor, Wen et al. (2011) indicated it is not particular critical to induce landslide, and their work was based on the data after the impounding of Three Gorges Project in 2003. The data source of our work is also after the impounding of Three Gorges Project, and we have stated that "It would be more convincing to use landslide data after 2006 to verify feasibility of the susceptibility assessment model" when we try to verify our results. We are sorry about our unclear description about temporal sequence and landslide influencing factors.

2. Data source and processing are not adequate. For quantitative assessment of landslide susceptibility, data is essential even the data source is always very difficult in a regional scale. Nevertheless, it is not tolerable in the paper that there are some unreasonable parameters considering the real soil/rock properties which significantly affect the reliability of calculation. Questions to table 1. (1) h, slope height, ranges 8-14.9? Normally, a natural slope is much higher as hundreds meters. But, in table 4 the selected range of h is higher than 100m. (2) c, cohesion, is presented in MPa which generally means a contact rock material not a soil? How this high cohesion is in relation to the shallow soil slope failures? (3) ÏŠ, weight, minimum is 13.7KN/m3, what kind of soil is it? The reviewer thinks the proper data source and processing are the first issue to ensure the reliability of the analysis.

We are grateful to the reviewer for pointing out our inadequacy and error of the data

source. As we know, a reliable data source is a basis to ensure the reliability of the analysis, which is a key component of the paper. Due to our carelessness, some unreasonable parameters were presented in the paper without rechecking. We have checked and corrected those inadequacies in our paper according to detailed field survey and related geotechnical experiments.

3. Quantitative calculation of FS in ASD seems not to be realistic. Questions on FS in table 2. From the reviewr's experience in FS calculation of safety factor, FS = 0.38 of group 15 slope is calculated with parameters of friction angle 34.14, cohesion 86.28Mpa, and 8.37 mm/day of rainfall. Is it corrected? But FS = 4.07 of group 9 slope is calculated with parameters of friction angle 5.31, cohesion 93.14 Mpa, and 29.30 mm/day of rainfall? These two FS values made me confused. Comparing to all FS values in Table 2, how to evaluate the reliability of FS? The reviewer thinks the training samples are the core for later susceptibility analysis. Please re-check your data source and your calculation.

We appreciate the reviewer for his/her rigorousness. As we mentioned above, the inadequacy and error of the data source may be a reason for the unrealistic calculation results. That we are so confident in quantitative calculation of FS in ASD without reconsidering is another reason. We will calculate the safety factors based on a new source data, and recheck the calculation results.

4. Are the key words present the key points of the paper? It should be re-considered?

The objective of the paper is rainfall-induced landslide susceptibility, therefore we choose it as the key word. Fengjie County is located in the Three Gorges region, known as an area of frequent landslides. Landslide hazards are increased in the Three Gorges area due to the construction of Three Gorges dam, that is the reason we choose a place located in Three Gorges region as our research target, and we think it could present the points of the paper. With regard to the key word 'Geo-studio', we decide to remove it from the key words after re-consideration, because the word 'Geo-studio' could not

present the key points of the paper.

5. Page 2 lines 43-44, "however, few landslides inventory maps: : :". In fact, very detailed landslide inventories, geological exploration, even many slope stabilization engineering have been already completed which is organized by central government.

Indeed, the central government of China has taken great efforts to deal with geo-hazards, inventory maps and landslide stabilization have been conducted very well. However, Three Gorges region is a large area, the government measures could not pay attention to all sides. Moreover, the landslide inventory of Fengjie county which is helpful to conduct a statistical analysis or heuristic analysis has no more than 20 years since the environment in Fengjie has experienced large changes after 2003.

6. Page 3 line 2, "hence the landslides events in the past may not be a good indication to implement landslide assessment". This statement is not geologically correct. The past is a good indication for future, especially for regional susceptibility mapping. We always use the inventory maps and recorded events as the data source which demonstrated the importance of the past.

We have checked and corrected our unreasonable statement, and the corrected statement should be 'hence the landslides events before 2003 may not be a good indication to implement landslide assessment'.

7. Page 4 line 42, "the dynamic factors are usually rainfall and earthquake". Reservoir water level variation is certainly an important dynamic factor which affects the bank slope in this paper. Reservoir water level plays two key roles to destabilize the bank slope: soften soil/rock properties and changing the pore water pressure. Why the paper does not use this parameter?

Reservoir water level variation is a dynamic factor which affects the bank slope, Wen et al. (2011) indicated it is not particular critical to induce landslide, and their work was based on the data after the impounding of Three Gorges Project in 2003.

8. Page 5 line 38 –40, "Bishop's simplified method and Morgenstern-price method…adopted". Which method is employed in the calculation?

In this paper, the Morgenstern-price method was adopted.

Reviewer#2: 1.The data and the results are not presented in a clear and well-structured way. 2. The method proposed by the Authors is not clear. In the Abstract they state the methodology is based on a "combination of mechanical stability analysis and artificial neuralnetwork (ANN) and of Geographic Information Systems (GIS ) and detailed field investigation".At the end of the introduction they state "The study develops an infinite stability model using Geo-studio software concerning rainfall infiltration to obtain safety factor for individual slopes, then combining the calculation results with artificial neural network (ANN) to figure out the relationship between influencing factors and potential landslides, based on the trained model, using GIS, a landslide susceptibility assessment map could be made". Yet they do not clearly explain, in chapter 3, how the various parts of the procedure (i.e. data from field investigation, stability analyses, ANN, susceptibility mapping) work and interact. The hypotheses of the various analyses performed are not stated.

It is really true as reviewer suggested that the data and the results are not presented in a clear and well-structured way and the method proposed by the authors is not clear. In view of the shortcomings of the paper in data preparation and processing. We will re-write this part according to the reviewer's suggestion.

3. Concerning the data from the field investigation, how are the ranges of values presented in Table 1 retrieved? How did the Authors derive the 30 combinations of values reported in Table 2? Where do the original data come from? Are they representative of the field conditions in the whole study area or only in the areas where landslides occurred? For instance, the unit weight and the strength properties (extremely large ranges are reported for the cohesion and friction angle values) are representative of what type of soil/rock? How may landslides are there in the study area and in what

type of soil/rock do they occur?

The field work was carried out in collaboration with the Institute of Exploration Technology in 2006, which covered landslides of the new urban area of Fengjie County, detailed information can be seen in the table below. We are sorry about our rough description about representative of soil/rock type, and the related description will be added in the paper. In the paper the ranges of values presented in Table 1 were retrieved according to detailed field survey and related geotechnical experiments, then the 30 combinations of values reported in Table 2 were derived by uniform design method as we presented in Page 5 line 19-23, "To build the ASD, one of experiment design methods, uniform design method was adopted to make the samples cover those 6 factors and guarantee sufficient experiment levels. The experiment levels of the factors were designed as 30, accordingly, an uniform design table U30*(3013) was utilized. Afterwards, the MATLAB software was employed to divide the range value of each factor into 30 levels uniformly, and then combine the divided levels of the factors together".

4. Concerning the stability analyses, are they conducted considering an infinite slope schematization (as written in page 3, line 6 and page 4, lines 19-20) or a boundary value problem in 2D (as one may infer from the fact they use the Geo-studio software)?In the latter case, what's the geometry of the considered slope(s)? The Authors do not provide any detail on the schematization of the slopes and on the shape and position of the sliding surfaces. What's the meaning of the 30 factors of safety computed (Table2) in relation to the slopes and landslides present in the study area?

We conducted stability analyses considering a boundary value problem in 2D, the geometry of the considered slope of group 12 is illustrated as below. As can be seen from fig.1, the schematization of the slopes and on the shape and position of the sliding surfaces can be clearly identified. The 30 safety factors were computed and utilized as training samples by ANN, then the trained ANN could be used to figure out the relationship between the slopes and potential landslides in the study area.

5. Concerning the ANN and susceptibility mapping, is the data from Table 3 (derived fromTable 2) the only data used to train the ANN looking for a relationship between the six so-called "influencing factors" and the safety factor? If so, what's the significance of the trained ANN in relation to the slopes and landslides present in the study area? How were the FS values computed with the trained ANN converted into the five reported classes (Table 6 does not report any range of values for FS)? What are the landslide data used for validation, years 1998-2014 or 2006-2014? In the paragraph starting at the end of page 7 the Authors state "after the impounding of Three Gorges Project in 2003, the environment in Fengjie has experienced large changes, thus the bank slopes would have a fair chance to slide owing to water-level rising .. It would be more convincing to use landslide data after 2006 to verify feasibility of the susceptibility assessment model." Does it mean the trained ANN is valid only for conditions in the study area successive to the impounding of Three Gorges Project in 2003? It appears that the Authors believe there's a role played by the reservoir level. If so, it should have been considered in the analyses.

We use the data from Table 3 to train the ANN because the 30 group samples are sufficient to reflect a relationship between six "influencing factors" and the safety factor. The significance lies in that the employment of ANN is a bridge between the individual slope stability analysis and the overall slope stability analyses in a regional scale since the uncompleted and biased database of rainfall intensity and duration, landslide magnitude and volume, slope failure patterns and landslide processes prevent a quantitative assessment of rainfall-induced landslide susceptibility. With regard to FS values classes, we will complete this in our paper. The landslide data used for validation is from Chongqing Institute of Geology and Mineral Resources, which ranges from 1998 to 2014, and we only took the landslides after 2006 to validate our results the trained because we believe the trained ANN is valid only for conditions in the study area successive to the impounding of Three Gorges Project in 2003. However, we do not believe the reservoir level play an important role in triggering a landslide. After the impounding of Three Gorges Project in 2003, the environment in Fengjie experienced

large changes in the very beginning. When it happens to landslide triggering, as shown in Table 8, landslides experience a sharp increase only in 2003.

6. Some of the sentences in the discussion session seem to disprove the relevance of the obtained results, e.g. "Our work did not take the water-level rising effect into consideration as there exist problems to quantify the effect. Hence it should be more careful when applied the rainfall-induced landslide susceptibility model in the study area. .. Due to the uncertainty lies in rainfall patterns and slope properties, it is difficult to precisely predict a landslide, and the slope failures may not in accordance with the predictions." The use of the English language is not adequate.

The sentences in the discussion session have been corrected according to the reviewer's suggestion.

Please also note the supplement to this comment:
https://www.nat-hazards-earth-syst-sci-discuss.net/nhess-2017-99/nhess-2017-99-AC1-supplement.pdf

[Figure]

**Fig. 1.**

| Landslide number | Name | Area (m²) | Volume (m³) |
|---|---|---|---|
| 1 | Houzishi | 12.19 | 450 |
| 2 | Liujiawan | 20.88 | 793 |
| 3 | Zhiwuyou | 2.83 | 100 |
| 4 | Laofangzi | 9.71 | 480 |
| 5 | Sichouchang | 19.3 | 1890 |
| 6 | Wangjiaping | 14 | 420 |
| 7 | Minjiabao | 62 | 3596 |
| 8 | Wolonggang | 4.7 | 126 |
| 9 | Xiangjiatang | 62.93 | 288.87 |
| 10 | Dahegou-Yuzhong | - | - |
| 11 | Andu | 2.04 | 49 |
| 12 | Chenjiabao | 9 | 200 |
| 13 | Chatupo | 3 | 33.2 |
| 14 | Tiehejinchang | 2.5 | 70 |
| 15 | Huangguaping | 2.6 | 240 |
| 16 | Laowuli | 3.7 | 40 |
| 17 | Shuitianba | 5.6 | 84 |
| 18 | Zhuangchang | 0.3572 | 1.7 |
| 19 | Liujiabao | 6.5 | 235 |
| 20 | Baotaping | 25 | 250 |
| 21 | Baiyian | 80 | 3609 |
| 22 | Yongle | 12.2 | 345 |
| 23 | Madaozi | 21.6 | 1017 |
| 24 | Yanjiapo | - | - |
| 25 | Linjiawan | 10 | 200 |
| 26 | Chenjiagou | - | - |

**Fig. 2.**